# Waypoint-Based Imitation Learning for Robotic Manipulation

Lucy Xiaoyang Shi[*]  Archit Sharma[*]  Tony Z. Zhao  Chelsea Finn

Department of Computer Science
Stanford University
{lucyshi,architsh,tonyzhao,cbfinn}@stanford.edu

**Abstract:** While imitation learning methods have seen a resurgent interest for robotic manipulation, the well-known problem of *compounding errors* continues to afflict behavioral cloning (BC). Waypoints can help address this problem by reducing the horizon of the learning problem for BC, and thus, the errors compounded over time. However, waypoint labeling is underspecified, and requires additional human supervision. Can we generate waypoints automatically without any additional human supervision? Our key insight is that if a trajectory segment can be approximated by linear motion, the endpoints can be used as waypoints. We propose *Automatic Waypoint Extraction* (AWE) for imitation learning, a preprocessing module to decompose a demonstration into a minimal set of waypoints which when interpolated linearly can approximate the trajectory up to a specified error threshold. AWE can be combined with any BC algorithm, and we find that AWE can increase the success rate of state-of-the-art algorithms by up to 25% in simulation and by 4-28% on real-world bimanual manipulation tasks, reducing the decision making horizon by up to a factor of 10. Videos and code are available at https://lucys0.github.io/awe/.

**Keywords:** imitation learning, waypoints, long-horizon

## 1 Introduction

The simple supervised learning approach of behavioral cloning (BC) has enabled a compelling set of robotic results, from self-driving vehicles [1] to manipulation [2, 3, 4, 5, 6]. However, due to the lack of corrective feedback, errors grow quadratically in the length of the episode for behavior cloning (BC) algorithms [7, 8], colloquially known as the *compounding errors* problem. Waypoints [9, 10, 4, 11] are a relevant proposition in this context: breaking the demonstration into a subset of states that can reconstruct the trajectory reduces the effective length of the decision-making problem, addressing the compounding er-

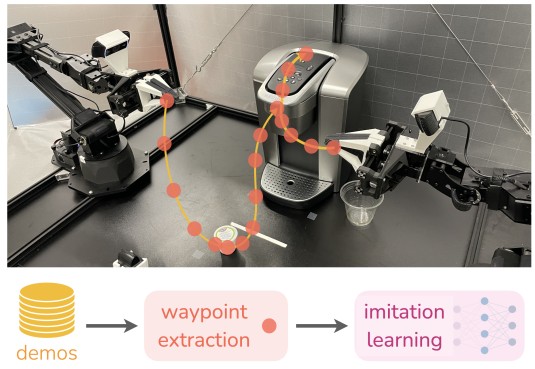

Figure 1: Our approach reduces the horizon of imitation learning by extracting waypoints from demonstrations.

rors problem while still allowing the use of simple methods such as BC. Our primary objective is to select a set of waypoints to reduce the effective horizon of the demonstration, and not necessarily find key bottleneck states. However, labeling waypoints is both an underspecified problem and requires additional human supervision.

---

[*]Equal contribution.

7th Conference on Robot Learning (CoRL 2023), Atlanta, USA.

Our objective is to develop a method for selecting waypoints for a given demonstration without any additional human supervision. For a robot arm, if a trajectory segment can be approximated linearly, a low-level controller can reliably imitate the segment without explicitly learning the intermediate states. Thus, we can represent that segment just by the endpoints. Extending this argument to arbitrary trajectories, we can use a subsequence of states as waypoints to represent the trajectory if the trajectory can be approximated well by linearly interpolating between the selected waypoints. The BC prediction problem then transforms from predicting the next action to the next waypoint.

How do we select the waypoint sequence that approximates a given a trajectory? Given a budget for the reconstruction error, where the error is defined as the *maximum* proprioceptive distance between the actual and reconstructed trajectory, we want to select the *shortest* subsequence of states for which the reconstruction error is within budget. This can be posed as a standard dynamic programming problem, by iteratively choosing an intermediate state as the waypoint and recursively selecting waypoints for the two resulting trajectory segments. The recursion terminates whenever the reconstruction error for the linearly interpolated trajectory between the endpoints is already within the budget. Importantly, finding the sequence of waypoints relies only on the robot's *proprioceptive* information, which is already collected during teleoperation. AWE makes no additional assumption about the extrinsic environment (state estimation, point clouds, etc.) and requires no additional label information from humans.

Overall, our work proposes *Automatic Waypoint Extraction* (AWE), a preprocessing module that breaks an expert demonstration into sequence of waypoints. AWE requires minimal additional information, and thus, can easily be plugged into current BC pipelines. We combine AWE with two state-of-the-art imitation learning methods, diffusion policy [5] and action-chunking with transformers (ACT) [6], and study its performance when learning from human-teleoperated demonstrations. On two existing simulated imitation learning benchmarks and multiple real bi-manual manipulation tasks, we find that AWE consistently improves performance, with up to 25% increase in success rate in simulation and 4-28% increase in success rate on real-world tasks.

## 2   Related Work

Imitation learning is a long-studied approach to training robotic control policies from demonstrations [1, 12, 13]. A central challenge is when compounding errors cause the policy to drift away from states seen in the demonstration data [8], leading to poor performance with small demonstration datasets. Prior approaches have aimed to improve imitation learning performance by developing new policy architectures [14, 11, 15, 16], using policies with expressive distribution classes [17, 18, 5], introducing modified action spaces [19, 20, 21, 22, 4], constructing modified training objectives [2, 6], utilizing particular visual representations [23, 24], incorporating data augmentation [25, 26], or collecting online data [8, 27, 28]. We instead aim to tackle the challenge of compounding errors by extracting waypoints that shorten the horizon. Our approach is orthogonal to and complementary to many of these prior developments; indeed, our experiments show that AWE can be combined with two recent, representative methods [5, 6] to improve their performance.

Prior works also attempted to reduce the policy horizon. Some use hand-defined high-level primitives [19, 22, 29, 30, 31, 32, 33, 34], but they lack flexibility and require extra engineering. Belkhale et al. [35] proposes a hybrid action space that incorporates both sparse waypoints and dense actions. While innovative, it requires humans to label waypoints either during data collection or post-processing, which may limit its scalability. Other recent works extract waypoints using various heuristics [9, 10, 4, 11], such as selecting waypoints at timesteps when the robot is at zero-velocity or actuating the gripper [4]. We are inspired by the success of these methods: they provide dramatic performance and data-efficiency improvements in some settings. However, we find that the heuristics do not apply in general, leading to low success on imitation learning benchmarks and fine manipulation tasks (Sec 5.4). In contrast to these methods, we circumvent the need for human waypoint labelling or heuristics. Our approach instead automatically extracts waypoints that minimize the trajectory reconstruction cost, which yields improvements on two existing simulated manipulation benchmarks and multiple real robot tasks. We discuss more related works in Appendix D.2.

# 3 Preliminaries

**Problem Setup**. We assume an expert collected dataset of demonstrations $\mathcal{D} = \{\tau_0, \tau_1 \dots \tau_n\}$, where each trajectory $\tau_i = \{(o_j, x_j)\}_{j=1}^{|\tau_i|}$ is a sequence of paired raw visual observations $o$ and proprioceptive information $x$. The proprioceptive information can either be the end-effector pose or joint pose, and includes the gripper width. In this work, we use pose-control for the action space, i.e., proprioceptives $x$ are the action outputs as well. Next, we briefly review two recent successful methods for BC, which we will use in our experiments.

**Diffusion Policy**. Diffusion policy [5] models the conditional action distribution as a denoising diffusion probabilistic model (DDPM) [36], allowing for better representation of the multi-modality in human-collected demonstrations. Specifically, diffusion policy uses DDPM to model the action sequence $p(A_t \mid o_t, x_t)$, where $A_t = \{a_t, \dots a_{t+C}\}$ represents a chunk of next $C$ actions. The final action is output of the following denoising process [37]:

$$A_t^{k-1} = \alpha \left( A_t^k - \gamma \epsilon_\theta(o_t, x_t, A_t^k, k) + \mathcal{N}(0, \sigma^2 I) \right), \tag{1}$$

where $A_t^k$ is the denoised action sequence at time $k$. Denoising starts from $\mathcal{A}_t^K$ sampled from Gaussian noise and is repeated till $k = 1$. In Eq 1, $(\alpha, \gamma, \sigma)$ are the parameters of the denoising process and $\epsilon_\theta$ is the score function trained using the MSE loss $\ell(\theta) = (\epsilon_k - \epsilon_\theta(o_t, x_t, A_t^k + \epsilon_k, k))^2$. The noise at step $k$ of the diffusion process, $\epsilon_k$, is sampled from a Gaussian of appropriate variance [36].

**Action Chunking with Transformers**. Action chunking with transformers (ACT) [6] models the policy distribution $p(A_t \mid o_t, x_t)$ as conditional VAE [38, 39], using a transformer based encoder and decoder. The decoder output is a chunk of actions of size $C$. Chunking is particularly important for high-frequency fine-grained manipulation tasks, with chunk sizes $C$ being as high as 100 [6].

# 4 Automatic Waypoint Extraction for Imitation Learning

The goal of this section is to develop our method for Automatic Waypoint Extraction (AWE). First, we define an objective that assesses the quality of the reconstructed trajectory for a given sequence of waypoints. Next, we show how a simple dynamic programming algorithm can be used to select the minimal number of waypoints that have a reconstruction error below a specified threshold. Finally, we discuss how to preprocess a demonstration dataset using AWE before plugging into the BC algorithm, along with the some practical considerations when training and evaluating a waypoint-based policy.

**Reconstruction Loss**. For an expert demonstration $\tau$, define the sequence of proprioceptives as $\tau_p = \{x_j\}_{j=0}^{|\tau|-1}$ and let $\mathcal{W}$ denote a sequence of waypoints such that $\mathcal{W} = \{w_0, \dots w_L\}$, where $w_i$ denotes the proprioceptive information in the waypoint. We reconstruct an approximate trajectory $\hat{\tau}$ by interpolating between the waypoints, i.e., $\hat{\tau} = f(\mathcal{W})$ for an interpolation function $f$. While we restrict to linear interpolation in this work, the framework can be extended to incorporate splines.

To measure how well a sequence of waypoints approximates the true trajectory, we measure how much the interpolated trajectory deviates from the true trajectory. We define the reconstruction loss as the maximum distance of any state in the original trajectory from the reconstructed trajectory, that is,

$$\mathcal{L}(\hat{\tau}, \tau) = \max_{x \in \tau_p} \min_{\hat{x} \in \hat{\tau}} \ell(x, \hat{x}) \tag{2}$$

where $\ell(\cdot, \cdot)$ is some distance function (for example, Euclidean $\ell_2$ distance). The $\min_{\hat{x} \in \hat{\tau}} \ell(\cdot, \hat{x})$ denotes shortest distance of a proprioceptive state to the interpolated trajectory $\hat{\tau}$. How do we aggregate projection errors for proprioceptives in the true trajectory? While there are several options, for example, mean projection error over $\tau_p$, we define the reconstruction loss as the maximum projection error over all proprioceptives in $\tau_p$. The success of a trajectory often relies on reaching key states, and the mean error can be low while having a high projection error for those key states. While a low reconstruction loss with maximum projection error also does not guarantee downstream success, it encourages minimizing outlier projection errors potentially critical for a successful execution. The reconstruction loss $\mathcal{L}$ is visualized in Figure 2.

**Waypoint Selection via Dynamic Programming**. Given the reconstruction loss, how do we use it to optimize waypoints? We consider the following optimization problem:

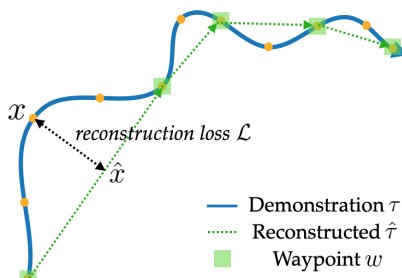

$$\min_{\mathcal{W}} |\mathcal{W}| \quad \text{s.t.} \quad \mathcal{L}(f(\mathcal{W}), \tau) \leq \eta, \qquad (3)$$

i.e., minimize the number of selected waypoints such that the reconstruction loss is below the budget $\eta$. As presented, waypoints can be arbitrary points in the proprioceptive space, but we will restrict waypoint selection to the states visited in the expert trajectory $\tau$. The problem simplifies

Figure 2: Visualizing the loss $\mathcal{L}$.

to finding the shortest subsequence of $\tau_p$ such that the reconstruction loss is less than $\eta$, which can be solved efficiently with dynamic programming (DP). For a trajectory segment, either linearly interpolating between the endpoints sufficiently reconstructs the segment (i.e, reconstruction loss less than $\eta$), in which case the endpoints are returned as waypoints. Or for every intermediate state between the endpoints: (1) break the trajectory into two segments at that intermediate state and, (2) recursively find the shortest subsequence for each segment. Finally, choose the intermediate state resulting in the shortest subsequence when the waypoints from its two trajectory segments are merged, and return the merged waypoints. The pseudocode for selecting waypoints with DP is in Algorithm 1.

**Preprocessing Demonstrations**. For an expert trajectory $\tau = \{(o_0, x_0), \ldots (o_T, x_T)\}$, denote the selected waypoints as $\mathcal{W} = \{(w_0, t_0) \ldots (w_L, t_L)\}$, where $w_i$ denotes the waypoint and $t_i$ denotes the time index in $\tau$. The training problem for a BC algorithm changes from predicting the next proprioceptive state to next waypoint. However, if done naïvely, the training dataset of next waypoints will be much smaller. But, we can use all observations in $\tau$ between two consecutive waypoints by labeling them with closest waypoint after the observation. This follows from the intuition that following the waypoints implies the robot tries to reach $w_{k+1}$ from $w_k$, and therefore, should target $w_{k+1}$ from intermediate states between them as well. The final dataset can be written as $\mathcal{D}_{\text{waypoint}}^{\tau} = \{(o_t, x_t, w_{\texttt{next\_wp}(t)})\}_{t=0}^{T-1}$ where $\texttt{next\_wp}(t) = \arg\min_{j \in \{0,1,\ldots L\}}$ such that $t_j > t$. The process of selecting waypoints and constructing the augmented dataset is repeated for every expert demonstration $\tau \in \mathcal{D}$, and the resulting datasets are merged to get the final training dataset.

**Overview and Practical Considerations**. We have proposed AWE, a simple method that can preprocess a demonstration into sequence of waypoints without any additional supervision. The training dataset can be relabeled with the next waypoint instead of next propriopceptive state, and plugged into a BC pipeline. The choice of policy distribution class used with AWE is important; waypoints introduce increased multi-modality into the conditional action distribution as different demonstrations may be processed into different waypoints. Using more expressive policy classes capable of representing multi-modal action distributions is critical, as introducing waypoints can make the performance *worse* for less expressive policy classes (Figure 6).

Why does AWE return meaningful waypoints? An intuitive notion of waypoints relies on registering important events happening in the extrinsic environment (grasping a cup, opening a door, etc.) while AWE uses just the proprioceptive information to select waypoints. AWE relies on the idea that the expert demonstrations will naturally deviate from linear motion during such key events. For simpler parts of the task, such as free-space reaching, demonstrations are more likely to be approximated by linear motion, resulting in fewer waypoints. Moreover, decreasing the budget $\eta$ allows for selecting more waypoints in general, and thus, better reconstruction as visualized in Figure 4.

An important consideration at test-time is to allow more time for position-control to reach waypoints, as waypoints are farther apart compared to proprioceptive positions in the original expert demonstrations. The exact instantiation for the low-level controller depends on the whether the robot is operating in the joint space or end-effector space, which we discuss in Appendix C.1.

## 5   Experiments

Our experiments seek to answer the following questions: (1) How well does AWE combine with representative behavioral cloning methods? (2) Can it be used to tackle standard imitation learning

Table 1: **Success rate (%) for simulated bimanual tasks.** We report results on both training with scripted data and training with human data, with 3 seeds and 50 policy evaluations each. Baseline results are obtained from Zhao et al. [6]. Overall, AWE +ACT significantly outperforms previous methods.

| | Cube Transfer | | Bimanual Insertion | |
|---|---|---|---|---|
| | scripted data | human data | scripted data | human data |
| BC-ConvMLP | 1 | 0 | 1 | 0 |
| BeT [14] | 27 | 1 | 3 | 0 |
| RT-1 [15] | 2 | 0 | 1 | 0 |
| VINN [24] | 3 | 0 | 1 | 0 |
| ACT [6] | 86 | 50 | 32 | 20 |
| AWE +ACT (Ours) | **99** | **71** | **57** | **30** |

benchmarks with real human demonstrations? (3) Can it be effective on a real-robot? (4) How does the parameterization of the policy affect the performance? (5) How do the selected waypoints and downstream performance change as we vary the hyperparameters? To answer these questions, we compare the performance of recent state-of-the-art BC methods with and without AWE on 8 tasks and 10 datasets. First, we evaluate AWE on a set of simulation environments, specifically two bimanual manipulation tasks from Zhao et al. [6] and three manipulation tasks from the RoboMimic benchmark [17]. We evaluate AWE on a set of three bimanual manipulation tasks on the real robot: *coffee making*, *wiping the table* and *screwdriver handover*. Hyperparameter and implementation details can be found in Appendix B and C respectively.

### 5.1 Bimanual Simulation Suite

The bimanual simulation suite contains two fine-grained long-horizon manipulation tasks in MuJoCo [40]. The observation space includes a $480 \times 640$ image and the current joint positions for both robots. The 14-dimensional action space corresponds to the target joint positions. Demonstrations are 400 to 500 steps at a control frequency of 50Hz. In the **Cube Transfer** task, the right robot arm needs to pick up the cube from a random position on the table, and then hand it to the left arm mid-air. For **Bimanual Insertion**, both the peg and the socket are placed randomly on the table. The arms need to first pick them up respectively, then insert the peg into the socket mid-air. Both tasks require delicate coordination between the two arms and closed-loop visual feedback: error in grasping can directly lead to failure of handover or insertion.

Two datasets are available for each task: one collected with a scripted policy, and one collected by human demonstrators, both with 50 demonstrations. As shown in Table 1, AWE outperforms competitive BC baselines on all tasks and datasets in the bimanual simulation suite, where some of the baselines completely fail due to task difficulty. AWE can increase the success rate of ACT, the state-of-the-art method on this benchmark 17% on an average, and up to 25% on the scripted bimanual insertion. The effective length of the training demonstrations reduce by a factor of $7\times$ to $10\times$, even allowing for improvements on human data which is fairly multi-modal to begin with. Notably, the performance improves by 50% when imitating human demonstrations on the bimanual insertion task. Overall, this suggests that the benefit from reducing the effective training horizon exceeds any potential downside from the increased multi-modality introduced by AWE.

### 5.2 RoboMimic Suite

Next, we evaluate on three simulated tasks from the **RoboMimic** [17] manipulation suite: **Lift** where the robot arm has to pick up a cube from the table, **Can** where the robots are required to pick up a soda can from a large bin and place it into a smaller target bin, and **Square** where robots are tasked to pick up a square nut and place it on a rod. It is challenging due to the high precision needed to pick up the handle and insert it into a tightly-fitted rod. Episodes start with randomly initialized object configurations. All environments return RGB observations and the action space is the 6DoF end-effector pose, with an additional degree for the gripper.

We combine AWE with the state-of-the-art method on this benchmark, Diffusion Policy [5]. Since diffusion policy achieves a near-perfect success rates on **Lift**, **Can**, and **Square** when training on a

Table 2: **Success rate (%) for behavior cloning benchmark, RoboMimic (Visual Policy).** AWE + Diffusion is more data-efficient than previous methods. We evaluate the policy every 100 epochs across the training, and report the average of the max performance across 3 training seeds and 30 different environment initial conditions (90 in total). Results on LSTM-GMM and IBC are obtained from Chi et al. [5] for comparison to more traditional methods. The performance scaling is visualized in Figure 8.

| Task | # Demos | AWE + Diffusion (Ours) | Diffusion | LSTM-GMM | IBC |
|---|---|---|---|---|---|
| **Lift** | 30 | $100.0 \pm 0.0$ | $100.0 \pm 0.0$ | - | - |
| | 50 | $100.0 \pm 0.0$ | $100.0 \pm 0.0$ | - | - |
| | 100 | $100.0 \pm 0.0$ | $100.0 \pm 0.0$ | - | - |
| | 200 | $100.0 \pm 0.0$ | $100.0 \pm 0.0$ | 96 | 73 |
| **Can** | 30 | $69.0 \pm 1.4$ | $61.0 \pm 5.9$ | - | - |
| | 50 | $85.7 \pm 1.9$ | $82.3 \pm 3.3$ | - | - |
| | 100 | $95.3 \pm 1.7$ | $93.3 \pm 0.0$ | - | - |
| | 200 | $96.7 \pm 0.9$ | $97.3 \pm 2.5$ | 88 | 1 |
| **Square** | 30 | $62.3 \pm 3.3$ | $44.3 \pm 6.1$ | - | - |
| | 50 | $67.0 \pm 2.9$ | $57.3 \pm 4.2$ | - | - |
| | 100 | $91.7 \pm 3.9$ | $82.0 \pm 7.0$ | - | - |
| | 200 | $94.7 \pm 3.9$ | $95.0 \pm 4.1$ | 59 | 0 |

dataset with 200 proficient-human demonstrations, we focus our evaluation on how the performance scales with the number of demonstrations, both with and without AWE. The results in Table 2 suggest that AWE consistently improves the performance of diffusion policy as the number of demonstrations is scaled from 30 to 200, while both of them outperform LSTM-GMM and implicit BC [41] with half the demonstration data, or even less. The improvements are larger when the number of demonstrations is smaller or the task is longer-horizon, for example, an 18% increase in the success rate when using 30 demonstrations on the **Square** task.

### 5.3 Real-World Bimanual Tasks

For real-robot evaluations, we use ALOHA [6], a low-cost open-source bimanual hardware setup. The setup consists of two leader arms and two follower arms, where the joint positions are synchronized between the leaders and followers during teleoperation. The observation space consists of RGB images from 4 cameras: two are mounted on the wrist of the follower robots, allowing for close-up views of objects for fine-manipulation, and the other two are mounted on the front and at the top respectively. The demonstration data consists of 4 camera streams and the joint positions for each robot at 50Hz. We refer readers to the original paper for more hardware details.

We experiment with three long-horizon tasks, each requiring precise coordination between the two arms, illustrated in Figure 3. For **Screwdriver Handover**, the right arm needs to pick up the screwdriver that is randomly initialized in a 15cm ×20cm rectangular region (#1) and hand it to the left arm mid-air (#2), followed by the left arm dropping it into the cup (#3). For **Wiping the Table**, a roll of paper towels is randomly placed in a 15cm ×10cm region. The opening of the roll always faces the right side, with naturally occurring variations in length and spacing. The left arm presses on the roll to prevent it from moving (#1), while the right arm tears off one segment of the paper towel (#2) and places it on a fixed location to absorb the spilled liquid (#3). For **Coffee Making**, a small coffee pod is randomized in a 15cm ×10cm region. The left arm needs to pick it up (#1), followed by the right arm opening the coffee machine (#2). The left arm then carefully inserts the coffee pod into the slot (#3), with the right arm closing the lid (#4). Next, the right arm grasps a transparent cup with upto 2cm randomization in the position and places it under the coffee outlet (#5).

The three tasks emphasize precision and coordination, and involve deformable or transparent objects that can be hard to perceive or simulate. For example, placing the coffee pod into the machine requires high precision. It is easy for the gripper or coffee pod to collide with the machine due to the small clearance. The screwdriver handover task emphasizes the coordination between two arms. Grasping the paper towel requires accurate perception of the deformable material, which also has low-contrast against itself. The gripper needs to move accurately so as to only grasp the opening but not collide with the roll and push it away.

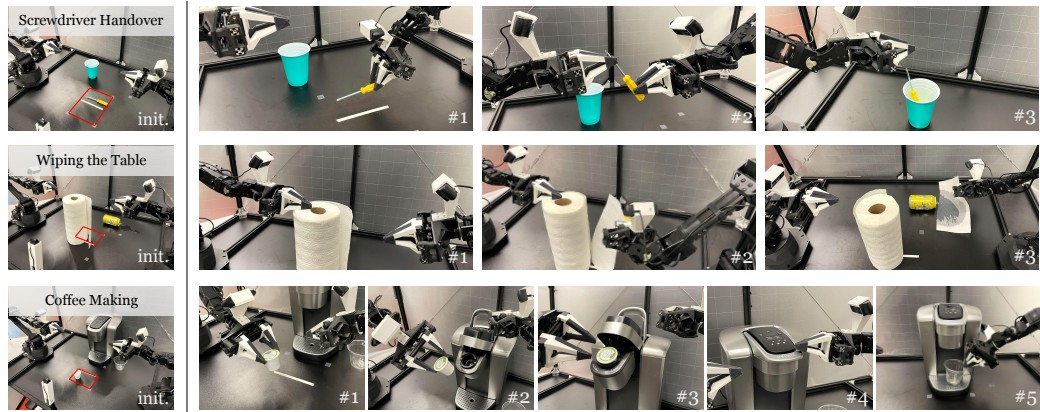

Figure 3: **Real-World Bimanual Tasks.** We consider three challenging real-world bi-manual tasks: (top) picking up a screw driver, handing it over to the other arm, and placing it in a cup, (middle) tearing off a segment of paper towel and putting it on a spill, and (bottom) putting a coffee pod into a coffee machine, closing the coffee machine, and placing a cup underneath the dispenser. Initial object positions are within the red rectangle.

Table 3: **Success rate (%) for real world tasks.** AWE improves the success of ACT on all three tasks ranging from 4% to 28%. On the longest horizon coffee making task, AWE improves success by 28%.

|                  | Screwdriver Handover | Wiping the Table | Coffee Making |
|------------------|:--------------------:|:----------------:|:-------------:|
| ACT              | 84                   | 92               | 36            |
| AWE + ACT (Ours) | **92**               | **96**           | **64**        |

As shown in Table 3, AWE achieves substantial success on each task. It consistently improves over ACT by 8%, 4%, and 28% on Screwdriver handover, Wiping table, and Coffee Making, respectively. We observe that the most common failure case for ACT is inaccurate action prediction, which results from compounding errors on these long-horizon tasks. For example, the robot may make a wrong prediction at the beginning and grasp the coffee pod at an inconvenient position. The subsequent predictions become increasingly incorrect, and thus the robot fails to insert the coffee pod into the machine. On the other hand, AWE can more accurately grasp the coffee pod due to a smaller decision horizon, resulting in more successful insertions into the coffee machine. Leveraging the low-level controller to execute linear motions instead of relying on accurate policy predictions can reduce the errors compounded over time. By accurately detecting waypoints for a successful handover, for tearing, and for inserting, AWE decreases the policy horizon and consistently improves performance.

### 5.4 Analysis

**Waypoint selection for different error budgets.** We visualize a ground truth trajectory of end-effector (EE) positions and the EE trajectory reconstructed using AWE for the **Can** task in Figure 4. As the the error budget $\eta$ is reduced, the reconstructed trajectory tracks the original trajectory better. Importantly, as the budget is decreased, waypoints are added to harder segments of the task, as they are less linear while the number of waypoints for simpler, linear paths stays similar. Smaller error thresholds lead to gradual increases in the number of selected waypoints. We also measure performance with varying error thresholds (the only hyperparameter), for AWE +DiffusionPolicy on the Can task with 50 demonstrations. Figure 5 shows that when the $\eta$ is too high (too few waypoints) or too low (too many waypoints), the agent does not take full advantage of AWE.

**On the importance of modeling multi-modality for AWE.** The usage of waypoints can increase the multimodality of the target conditional action distribution. We compare the performance of AWE when trained with mean-squared error (MSE) loss (i.e, a unimodal Gaussian with identity covariance) and a more expressive Gaussian mixture model (GMM) with 5 modes. As shown in Figure 6, GMM policies can benefit from AWE, as they can represent multimodal action distributions. However, vanilla BC with a MSE loss degrades in performance. BC has a mode-covering behavior, and insufficient representative power of unimodal Gaussian can cause the performance to degrade.

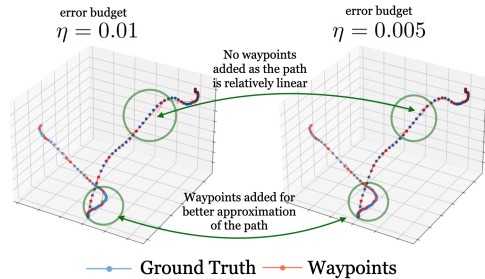

Figure 4: As the error budget $\eta$ decreases, our method selects fewer waypoints if linear interpolation aptly approximates the segment. Best viewed on our website.

Figure 5: Success rate vs. error budget threshold $\eta$. Performance drops slightly if the budget is too tight and more significantly if the budget is too permissive.

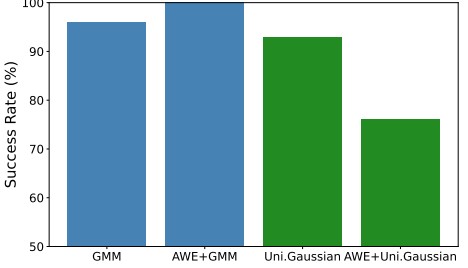

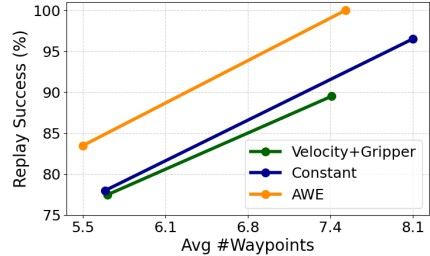

Figure 6: AWE requires expressive policy classes. While expressive policies that can represent multimodal distributions benefit from AWE (GMMs on *left*), the performance can degrade for policy classes that are not sufficiently expressive (*right*).

Figure 7: Comparing the **replay** success rate of AWE to common heuristics for waypoint selection. With similar numbers of waypoints, following waypoints from AWE leads to more consistent task completion than following the waypoints from heuristics.

**Comparison to heuristics.** Prior works [9, 10, 4, 11] have been successful by extracting waypoints using simple heuristics. Are simple heuristics enough for extracting important waypoints? We experiment with two heuristics. The first one is similar to Shridhar et al. [4], labeling timesteps as waypoints when the end-effector velocity is close to zero, or when the binary gripper state changes. The second heuristic selects waypoints with fixed intervals. For AWE and both heuristics, we extract waypoints for all 200 trajectories in the RoboMimic **Lift** dataset, and measure the success rate when replaying the demonstration, i.e., following the extracted waypoints starting from the demonstration trajectory's initial state. We adjust the selection threshold or interval to generate similar numbers of waypoints across methods for comparable results. Results in Figure 7 show that these two heuristics do not lead to satisfactory success rates even when simply replaying the trajectories.

## 6 Conclusion

We presented a method for extracting waypoints from demonstrations of robotic manipulation tasks, therefore reducing the horizon of imitation learning problems. We found that AWE can be combined with state-of-the-art imitation learning methods such as diffusion policy and ACT to improve performance, especially in data limited settings. AWE also consistently improved performance on three real-world dexterous manipulation tasks. Finally our analysis indicated the importance of the AWE optimization compared to naive or heuristic waypoint selection methods, as well as the effect of the error budget and policy distribution class on performance.

**Limitations.** AWE leverages proprioceptive information to reparameterize demonstration trajectories in end-effector or joint space, an approach that may not be applicable to torque-controlled robot arms, tasks requiring forceful manipulation, or other robotics problems such as purely visual navigation or legged locomotion. Our evaluation only considers quasi-static tasks, and AWE currently does not account for velocities, which might be important for dynamic tasks. Furthermore, for tasks that require extreme precision at certain times, we expect that AWE would require a tight error budget, diluting the benefit of using waypoints. This limitation might be resolved by identifying when such precision is needed, either automatically or by incorporating some human supervision, and subsequently modifying the AWE optimization objective.

**Acknowledgments**

This work was supported by Schmidt Futures and ONR grants N00014-20-1-2675 and N00014-21-1-2685. We would like to thank Suneel Belkhale and Chen Wang for helpful discussions, and all members of the IRIS lab for constructive feedback.

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

# A AWE Pseudocode

We provide the complete pseudocode for AWE in Algorithm 1.

---

**Algorithm 1** Automatic Waypoint Extraction (AWE)

---

**input**: $\mathcal{D}$; // expert demonstrations
**input**: $\mathcal{L}, f, \eta$;
// waypoint selection via dynamic programming
def *get_waypoints*$(\tau, \eta, \mathcal{M})$:
    **if** $\tau \notin \mathcal{M}$ **then**
        // check if the endpoints are valid waypoints
        **if** $\mathcal{L}(f(\{\tau.\mathtt{start}, \tau.\mathtt{end}\}), \tau) \leq \eta$ **then**
            $\mathcal{M}[\tau] = \{\tau.\mathtt{start}, \tau.\mathtt{end}\}$;
        // try all intermediate states as waypoints, and return the smallest set
        **else**
            // initialize length of current shortest subsequence
            $m \leftarrow \infty$;
            // loop over all intermediate states as waypoints
            **for** $w \in \tau.mid$ **do**
                $\mathcal{W}_{\text{before}} \leftarrow$ *get_waypoints*$(\tau.\mathtt{before}(w), \eta)$;
                $\mathcal{W}_{\text{after}} \leftarrow$ *get_waypoints*$(\tau.\mathtt{after}(w), \eta)$;
                // dedupe $w$, as it is in both of them
                $\mathcal{W} \leftarrow (\mathcal{W}_{\text{before}} \backslash \{w\}) \cup \mathcal{W}_{\text{after}}$;
                **if** $|\mathcal{W}| < m$ **then**
                    $m \leftarrow |\mathcal{W}|$;
                    $\mathcal{M}[\tau] \leftarrow \mathcal{W}$;
    return $\mathcal{M}[\tau]$;

// construct dataset for next waypoint prediction
def *preprocess_traj*$(\mathcal{W}, \tau)$:
    $\mathcal{D}_{\text{aug}} \leftarrow \{\}$;
    **for** $(o_t, x_t) \in \tau$ **do**
        // select the nearest future waypoint in $\mathcal{W}$
        $w \leftarrow \mathcal{W}.\mathtt{next\_waypoint}(t)$;
        $\mathcal{D}_{\text{aug}} \leftarrow \mathcal{D}_{\text{aug}} \cup \{(o_t, x_t, w)\}$;
    return $\mathcal{D}_{\text{aug}}$;

$\mathcal{D}_{\text{new}} \leftarrow \{\}$;
**for** $\tau \in \mathcal{D}$ **do**
    $\mathcal{M} \leftarrow \{\}$; // memoize waypoints for efficient dynamic programming
    $\mathcal{D}_{\text{new}} \leftarrow \mathcal{D}_{\text{new}} \cup$ *preprocess_traj*$(get\_waypoints(\tau, \eta, \mathcal{M}), \tau)$
**output**: $\mathcal{D}_{\text{new}}$

---

# B Hyperparameters

## B.1 Error Budget Threshold

The only hyperparameter we need for waypoint selection is $\eta$, the error threshold (Table 4). The threshold $\eta$ is the same for all data sizes $\{30, 50, 100, 200\}$ across all tasks on RoboMimic, i.e. $\eta = 0.005$. We also use a consistent $\eta$ for both scripted data and human data on both tasks in the Bimanual Manipulation suite, i.e. $\eta = 0.01$. Two out of three real-world tasks also use the same $\eta$; however, on the **Coffee Making** task, we opt for a lower $\eta$ to select more waypoints due to the high-precision nature of the task.

Table 4: Hyperparameter for waypoint selection.

| Task | Error thresholod ($\eta$) |
|---|---|
| Lift | 0.005 |
| Can | 0.005 |
| Square | 0.005 |
| Cube Transfer | 0.01 |
| Bimanual Insertion | 0.01 |
| Screwdriver Handover | 0.01 |
| Wiping Table | 0.01 |
| Coffee Making | 0.008 |

## B.2 ACT in Bimanual Simulation Suite

We use the same hyperparameters as the ACT paper [6], shown in Table 5, except reducing the chunk size from 100 to 50. Intuitively, as the length of trajectories reduces after running AWE, the chunk size can also be reduced to represent the same wall-clock time.

| Hyperparameter | ACT | AWE + ACT |
|---|---|---|
| learning rate | 1e-5 | 1e-5 |
| batch size | 8 | 8 |
| # encoder layers | 4 | 4 |
| # decoder layers | 7 | 7 |
| feedforward dimension | 3200 | 3200 |
| hidden dimension | 512 | 512 |
| # heads | 8 | 8 |
| chunk size | 100 | **50** |
| beta | 10 | 10 |
| dropout | 0.1 | 0.1 |

Table 5: Hyperparameters of AWE +ACT and ACT. The only difference is the reduction in chunk size.

## B.3 Diffusion Policy in RoboMimic

We use the exact same set of training hyperparameters as Diffusion Policy [5] (Table 6). The only additional hyperparameter we added is the "control multiplier" (bottom row), which allows the low-level controller to take more steps to reach the target position at the inference time. This can be useful when predicted waypoints are far apart.

## B.4 A Guide to Hyperparameter Selection

We suggest selecting an error threshold for new tasks based on a ratio of the number of waypoints to the average length of the trajectories. Our recommendation is to aim for a ratio of approximately 1:8, which can be automatically calculated using the waypoint generation script in our codebase. The ideal ratio may vary depending on the specific task and control frequency. Based on our empirical findings, a ratio between 1:5 and 1:15 tends to effectively reduce the policy horizon while still maintaining an accurate approximation of the trajectories.

For tasks involving real robots using ALOHA hardware [6], we advise turning on temporal ensembling (Sec C.3) to ensure smoother actions. Nonetheless, if the policy appears overly hesitant, two potential remedies are: (a) disabling temporal ensembling, and (b) increasing DT to emulate a blocking controller, where DT refers to the time interval between each update in a simulation or a control loop.

| Hyperparameter | Lift | Can | Square |
|---|---|---|---|
| Ctrl | Pos | Pos | Pos |
| $T_o$ | 2 | 2 | 2 |
| $T_a$ | 8 | 8 | 8 |
| $T_p$ | 10 | 10 | 10 |
| # $D$-params | 9 | 9 | 9 |
| # $V$-params | 22 | 22 | 22 |
| # Layers | 8 | 8 | 8 |
| Emb Dim | 256 | 256 | 256 |
| Attn Dropout | 0.3 | 0.3 | 0.3 |
| Lr | 1e-4 | 1e-4 | 1e-4 |
| WDecay | 1e-3 | 1e-3 | 1e-3 |
| $D$-Iters Train | 100 | 100 | 100 |
| $D$-Iters Eval | 100 | 100 | 100 |
| Control Multiplier | 10 | 1 | 10 |

Table 6: Hyperparameters for diffusion policy. Ctrl: position or velocity control, $T_o$: observation horizon, $T_a$: action horizon, $T_p$: action prediction horizon , #$D$-Params: diffusion network number of parameters in millions, #$V$-Params: vision encoder number of parameters in millions, Emb Dim: transformer token embedding dimension, Attn Dropout: transformer attention dropout probability, Lr: learining rate, WDecay: weight decay (for transformer only), $D$-Iters, Train: number of training diffusion iterations, $D$-Iters Eval: number of inference diffusion iterations, Control Multiplier: multiplier for the low-level control steps.

# C   Implementation and Experiment Details

## C.1   Controller

We use an Operation Space Controller (OSC) in RoboMimic, which allows position and orientation control of the robot's end-effector. It takes in the desired absolute position and orientation of the end-effector, and computes the necessary torques and velocities.

We use the default joint position controller in the Bimanual Manipulation suite. On real-world tasks, we made no change to the controller except for the **Coffee Making** task, where we increased the step time from 0.02 to 0.1. This allows the controller to operate closer to a blocking controller, and execute low-level actions longer until reaching the desired joint position.

## C.2   Loss Function

To determine the distance between potential waypoints and the ground truth trajectory, we project the ground truth state onto the linearly interpolated waypoint trajectory and compute the L2 distance for xyz position. For orientation, we convert the axis angles to quaternions and slerp two ground truth quaternions to determine the projection. Then we sum the position and orientation distances as the state loss. For the trajectory loss, we take a max over all states.

## C.3   Temporal Ensemble

For all the ACT experiments, we adopt a *temporal ensemble* technique as in the original paper [6]. Temporal ensembling is an approach to improve the smoothness of action chunking in robotic tasks. It queries the policy at each timestep, creating overlapping chunks and multiple predicted actions for each timestep. These predictions are then combined via a weighted average using an exponential weighting scheme, $w_i = \exp(-m \times i)$, that helps in smoothly incorporating new observations. This technique enhances the precision and smoothness of motion without any additional training cost, but requires extra computation during inference. We refer readers to Zhao et al. [6] for more details.

## C.4   Computation Cost

Computing waypoints is inexpensive, especially compared to the training budget. The wall clock time for labeling one trajectory in **Lift** is 0.8 seconds on average.

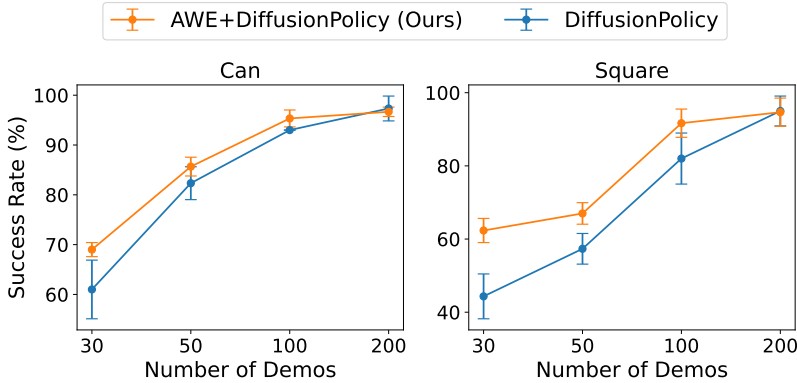

Figure 8: **Performance scaling with demonstrations.** We compare how the performance scale for diffusion policy [5] with and without AWE. Training on waypoints generated by AWE consistently improves the performance, with improvements being larger on the harder task (Square).

## D Additional Comparisons

### D.1 Subsampling

A potential alternative to our proposed AWE method is the straightforward approach of subsampling trajectories. This implicit selection of waypoints can be viewed as a heuristic method. Figure 7 demonstrates that AWE achieves a superior replay success rate, i.e., when one follows the extracted waypoints, starting from the demonstration trajectory's initial state, than subsampling. However, how do these methods influence the performance downstream?

To address this, we compare the success rate of a policy learned using waypoints selected by AWE against those from subsampled trajectories. We experiment on two RoboMimic tasks, Can and Square, both using 100 demonstrations. We compare against two subsampling ratios, specifically 5 and 7: a ratio of 7 produces a number of waypoints comparable to that of AWE in RoboMimic.

|  | AWE (Ours) | Subsampled by 7 | Subsampled by 5 |
|---|---|---|---|
| **Can (100 demo)** | 95.3 | 77.3 | 72.7 |
| **Square (100 demo)** | 91.7 | 77.3 | 86.4 |

Table 7: Comparison of success rates for policies learned using AWE and subsampling methods on Can and Square.

As shown in Table 7, we find that: 1) AWE consistently surpasses the subsampling approach; 2) The most effective subsampling ratio is task-dependent. For instance, in the Can task, subsampling by 7 exceeds subsampling by 5. Conversely, the Square task sees better results with a ratio of 5. This variance suggests that AWE can discern and select waypoints that are more instrumental for downstream learning.

### D.2 Keypose-based motion planning

Our work is conceptually related to keypose-based motion planning, which has seen notable contributions in recent years. Tonneau et al. [42] utilized keyposes for multi-robot planning, emphasizing bottleneck states. Ichter et al. [43] employed probabilistic roadmaps to identify critical configurations, while Lai [44] RRF* method adaptively targets bottleneck regions. These methods require complete knowledge of the environment to plan trajectories. In contrast, our method derives waypoints from the robot's proprioceptive data. We assume we only have raw RGB images and no low-level information about the external environment.

