# OpenReview forum: "Waypoint-Based Imitation Learning for Robotic Manipulation"
_robot-learning.org/CoRL/2023/Conference — CoRL 2023 Poster_

### Official Review · Reviewer_3s7U · 2023-07-14

**Confidence:** 5
**Originality:** Good
**Technical Quality:** Excellent
**Clarity Of Presentation:** Excellent
**Impact:** 3

**Recommendation:**

Weak Accept: I recommend accepting the paper, but will not argue for my recommendation if the majority of other reviewers have a different opinion.

**Review:**

The paper is really well written. From the abstract you can understand the problem and the idea of the proposed approach. And I have no comments about the presentation of the work, or organization of the manuscript. I only have minor comments on the textual part which follows next

1. in line 88, the subscript j is not defined.
2. citation [1] in line 64 does not see correct. Please double check.
3. In the label of table 3 and the claims, it says AWE improves success by 28%, it should be "from 4%, up to 28% in the tested cases", otherwise it is miss leading.
4. Figure 5, missing units in the x-axis, seems to be in meters?
5. In lines 183 and 184, the authors says that they compare AWE on 8 tasks and 10 datasets, but I do not know where this comparison is. Is that one of the tables in the text? please make it explicit.
6. line 146, mentions Algorithm 1, which is not to be found in the paper. Where is Algorithm 1? please make it explicit.

The biggest critic that I had about the approach is that the selection of way-points clearly creates ambiguities in the trajectories, which greatly limits the BC learning capabilities. We can clearly see that this situation would arise in Figure 1. Nevertheless, the authors propose a time-based selection (preprocessing demonstration, line 147) and direct the proposed method to be used with multi-modal models (On the importance of multi-modality for AWE, line 275), which are designed to solve such problems. This is correct and well done.
However, multi-modal models are harder to train, and an evaluation comparing how much data we need to train a: 1) a standard BC method without the way-points selection, 2) a multi-model BC without the way-points selection, and 3) a BC with the way-points selection would allows to evaluate the impacts of the selection and its synergy with multi-modal methods more complete than the current evaluation presented in Figure 6.

Still on the topic of ambiguities created by AWE, we have two mechanisms for handling them, the time-base selection and the multi-modal BC methods. It would be nice to have an evaluation for the contributions of each one of these individually, so we can understand their importance. From figure 6 I imagine that multi-modality is much more important, but we cannot asses the importance of the time-based selection. Thus, do we actually need both? Only multi-modal models are enough?

I think the authors might have missed one of the greatest contribution of Section 4. The proposed loss is actually forms a tighter than the mean projection error, and as such, we can see the later as a relaxation of the former, and the former describes better the actual objective. Maybe the wording "relaxation" is misused here, but I hope the parallel with standard optimization problems makes it clear. It would be enriching to elaborate on that discussion, as this seems to be a contribution applicable to several other works as well, and hence a strong contribution.

Nevertheless, the main limitation of the paper seems to be its application for trajectories that can be adequately  approximated by straight lines. This is not the case for quaternions, or joint states which are often in radians, the linear approximation in those cases can be mediocre at its best (this problem is probably mitigated by the learning method but we have no evaluation on that). These limitations can be addressed using higher degree polynomials (not only limited to splines as noted in line 117), or other regression methods, e.g., sine and cosine as basis functions for the regression, or even a NN. Of course, this is a topic for future works/journal extension.

On the results and analysis there are a few problems, which greatly hurt the claims of the paper:
1. On table 1, and 3 we only have the success rate in %, whereas we need a more complete statistical analysis, and the variances. Like in Table 2.
2. In table 2, the results for AWE + diffusion and diffusion lie within the variance of each other. As such we cannot conclude the  adding AWE makes things better as claimed by the authors. Please, restrict the claims to what the data shows. This also can make one doubt about the conclusions of tables 1 and 3.
3. In figure 5, please present the results for more points in the x axis, only 3 points are not enough for concluding the claims and that the behavior of the curve is what we expect.
4. Selecting a few points makes as the vector fields generated by the policy to have a "coarser" shape, basically what is happening is a non uniform weighting of the the velocity profiles of the model's vector fields. As such, we need to check the impact in the vector fields with and without AWE. This can provide insights or reveal problems which are hidden in the current state of the paper. Please add figures for the vector fields for a couple of cases, and FOR ALL cases in the complementary material.
5. The link to the source code in the paper's website is broken. Please provide the code, data, and instructions for installing and running, so we can review and understand better what is happening.

A final remark, in section 5.4, line 266, I do believe that to make the proposed approach complete, the authors should elaborate and study a method for defining the error threshold, making the contribution much more solid! In the state we are now, we do not even know how hard is to tune the only hyper-parameter in the method. Again, this is topic for future-works/journal extension.




**Quality Of The Limitations Section:**

Limitations are addressed clearly

**Questions For Rebuttal:**

I have no questions for rebutal, but please address the problems mentioned above. Fell free to rebut comments in case of disagreement.

**Robotics Focus:**

Sufficient demonstration on hardware

**Summary Of Paper:**

This paper tackle compound errors in Behavioral Cloning (BC) within the Learning from Demonstrations framework. The main idea is to sub-sample the demonstrations, reducing the number of points computed during rollouts, and hence reducing the errors. This creates the problem of selecting the demonstration subset. Selecting few points leads to loosing information, and too many points defeat the purpose of the method.

The main contribution of the paper is a simple, and yet effective, algorithm for selecting the subset of points based on approximating the demonstrated trajectories with straight lines, respecting a user given reconstruction error threshold, a.k.a, a hyper-parameter which controls the number of points and thus needs to be tuned for avoiding too many of too few points and the problems associated with those cases.

The method is a preprocessing step, and thus it is complementary to BC methods. The method is sufficiently tested, presenting promising results in most cases.

**Summary Of Recommendation:**

Please elaborate on the limitation listed in the review.

I cannot only recommend a strong accept because the results are somehow incomplete and the all the claims are not really supported by the results. A better analysis needs to be done, despite the results seem promising.

On the Impact, the proposed method generates ambiguities and the requirement of multi-modal models limits its applicability, and thus we cannot consider major impacts in robotics.

Being the problems with the results correctly addressed, I vote in favor of the publication of the paper.

---

> ### Author Response · Authors · 2023-08-15
> **Any Further Clarifications?**
>
> Thank you once again for the constructive feedback! Please let us know if our response addresses your questions and whether there is additional clarification we can provide.

---

### Official Review · Reviewer_cGVi · 2023-07-20

**Confidence:** 5
**Originality:** Fair
**Technical Quality:** Fair
**Clarity Of Presentation:** Good
**Impact:** 2

**Recommendation:**

Weak Reject: I recommend rejecting the paper, but will not argue for my recommendation if the majority of other reviewers have a different opinion.

**Review:**

**Pros:** The paper is easy to read and seems to tackle an interesting and challenging problem. However, the execution leaves many things to be desired.

**Cons:** There is very limited novelty in this paper. The idea of using waypoints is not very novel.

Infact, many approaches in robotics use an implicit form of this approach already. For instance, when we subsample trajectories that have been recorded at a high frequency we (i.e. reduce 100Hz trajectories to 10 Hz), we implicitly select waypoints and use linear interpolation between these waypoints. This is basically what the current paper proposes although they use a dynamic programming formulation to find these waypoints.

The same approach can also be seen as a slight generalization of the keypose based approaches which identify bottleneck states and then use planners to reach these bottleneck states. Infact, the idea of pre-processing demonstrations to add actions to next waypoint has been already proposed and extensively used in the keypose literature. Thus, overall, the only sort of novelty this paper has is to use a dynamic-programming approach to find waypoints.

Additionally, I don’t think using dynamic programming (DP) **separately** for each trajectory to find waypoints is optimal. First, since DP is being run separately for each trajectory, the waypoints that the algorithm ends up selecting may not be the consistent across trajectories. This inconsistency is quite sub-optimal since the idea of waypoints (as noted in the paper itself) is to find key bottleneck states, that are shared across demonstrations. But if the preprocessing selects very different waypoints the network will have to learn a much more complex policy which will generalize well.

Finally, for most tasks considered in the paper there is clearly an object-reach stage (e.g. object, pick (lift), object pick-place (can)), so why shouldn’t a waypoint that is close to the object state (this can be discovered directly from contact mode) be directly used. In my view, such a policy can work very well, it would be interesting to see how such a baseline acts on some of tasks considered here. Also, I don’t think the DP based approach will select such an object-centric frame at all (given 1 waypoint to select) since it’s just doing local curve fitting, but clearly we know that using a planner or even classical movement primitives for this object-centric state is sufficient for pick-place tasks.

**Quality Of The Limitations Section:**

Additional details required

**Questions For Rebuttal:**

please see above.

**Robotics Focus:**

Sufficient demonstration on hardware

**Summary Of Paper:**

This work proposes a waypoint based approach for learning manipulation tasks from expert demonstrations. To extract these waypoints a simple dynamic programming approach is proposed. The dynamic programming based approach tries to find waypoints that are temporally distant and also linearly connected.

**Summary Of Recommendation:**

The paper aims to tackle the challenging problem of covariate shift. However, in my opinion, the overall novelty is very limited. More importantly, the overall approach is also very arbitrary and not principled or grounded since there is no consistency across demonstrations and I think it's success heavily relies on task domains and kind of demonstrations being used for training.

---

> ### Author Response · Authors · 2023-08-15
> **Any Further Clarifications?**
>
> Thank you once again for the constructive feedback! Please let us know if our response addresses your questions and whether there is any additional clarification we can provide.

---

### Official Review · Reviewer_ZvMK · 2023-07-24

**Confidence:** 4
**Originality:** Good
**Technical Quality:** Very Good
**Clarity Of Presentation:** Good
**Impact:** 3

**Recommendation:**

Weak Accept: I recommend accepting the paper, but will not argue for my recommendation if the majority of other reviewers have a different opinion.

**Review:**

# Clarity

The paper is written fairly well and clear. Following remarks might help to improve the clarity of the paper.

## Minor remarks
- Figure 4: hard to see, especally in a printed version. Maybe a 2d projection would be better for illustration purposes
- mathematical notations are a bit confusing at times, e.g., line 114 vs line 88. Maybe better someting like $X_\tau$ instead of $\tau_p$.
- line 162: "additional multi-modality" => "additional modality"
- line 173: "test-time" => "run time"
- line 90: it is unclear which propriceptive data is actually being used in the presented experiments - joint angles or end the effector pose?
- line 146: "Algorithm 1" - referenced, but doesn't exist in the paper (is located in the Appendix).
- line 267: "EE trajectory" was not introduced ("end-effector trajectory"?)
- line 288, 116: ", i.e. " => ", i.e.,"

- Algorithm 1 (in the suplimentary material):
  * $get waypoints(\tau.before(w),\eta)$ => $get waypoints(\tau.before(w),\eta,\mathcal{M})$
  * $get waypoints(\tau.after(w),\eta)$ => $get waypoints(\tau.after(w),\eta,\mathcal{M})$

## Form
- paper makes extensive use of rethorical questions, which remain "unanswered" and reduce the clarity of the paper. They can be easyly reduced with concise statements. It would also save some space ;)
  * Example 1: (line 6-7) "Can we generate waypoints automatically without any additional human supervision?" - rhetorically, this needs to be closed with something like "We belive - yes, it is possible with our proposed method."
  * Example 2: (line 166) a suggestive rethorical question "Why does AWE return meaningful waypoints?". Only an intuitive explanation is provided, which is alright in itself, however the question remains open. Replace it with a straigt forward question or title, e.g.,
"Does AWE return meaningful waypoints?"
- it would be good to include the main algorithm $get_waypoints(...)$ into the main paper. Du reduce its size, the comments can be removed as there is suficient explanation in text already. This would make the paper cearer.


# Quality

Overall, the quality of the presented work is fairly good.

The empirical study consists of primary experiments (in simuation and on the robots) and a number of secondary experiments studying the limits of the proposed approach.
The primary experiments are comprehensive and demonstrate an improvement in performance when the proposed pre-processing method is used.


## Interpolation
lines 173-176: "allow more time" - is a wague statement.

For comparison of the performance between waypoint-trajectory and not-approximated trajectories it could be crucial, that the time between the waypoints is calculated based on the real time of the trajectory.
From the paper it is unclear if the overall action was executed with comparable speed.
For instance, executing the waypoint-trajectory significantly slower than original trajectory might affect the success of the action, e.g., reduce vibrations and make it more accurate.

How do other methods for waypoint selection handle interpolation?
Adding measured times for the execution of the action could also help (in case the learned actions vary in time).


## Linear / Constant

Although the way points are calculated by means of piece-wise **linear** approximation, the resulting augmented trajectory seems to piece-wise **constant**.

They waypoints are selected from the trajectory points such that the distance error between the piecewise linear continuous approximation and the trajectory is below a given threashhold.
The data is augmented with the newly calculated waypoints.  Thereby, the number of waypoint is smaller than the number of points in the trajectory itself.
The augmentation is done by extending each tuple in the trajectory by the closest waypoint. This results, however, in a piecewise constant approximation of the proprioceptive trajectory.

For instance, consider the following sequence:
{(o0, x0),     (o1, x1),     (o2, x2),     (o3, x3),     (o4, x4),     (o5, x5),     (o6, x6),     (o7, x7)}
Assume, that (x_i) is approximated throgh two waypoint (a single line). Then the augmented data could look like this:
{(o0, x0, w0), (o1, x1, w0), (o2, x2, w0), (o3, x3, w0), (o4, x4, w1), (o5, x5, w1), (o6, x6, w1), (o7, x7, w1)}
Meaning, that the sequence (x_i) was approximated through two constant functions: f1(i) = w0 and f2(i) = w1.

In principle it makes sense. However, from the paper it is not clear that the authors are aware of this fact as the algorithm is
formulated for piece-wise linear functions.
Perhaps, if the approximation is formulated from the beginning as being piece-wise constant,
the norm $l(.,.)$ in (line 124) could chosen to be more simple and easie to calculate.

As one of practical considerations, (line 173-176), the authors state that the actual iterpolation is done by the motor
controller resulting in a piece-wise linear trajectory.
However, the learning algorithms is presented with a picewise constant trajectory, which could be a signifficant aspect.


## Remarks on "seconday experiments"
Following the aspects could contribute to a more conclusive and substantiated analysis:

- Figure 7: how many waypoints were calculated in each instance? This would help to assess how the methods relate to each other.
- Table 2: the performance of "AWE+Diffusion" gets worse for "# Demos" = 200. This is not adressed - would be important to explore further to understand the lmitations of the algorithm.
- Figure 5: the effect of the error threashold is illustrated for "Can task with 50 demonstrations" (line 273).
The performance seems to decrease outside of a specific value. Is this value different for each task? How does it change with the number of demonstrations?
- how were the heuristic methods selected for comparison?

## Reduction in "compounding error"

With the proposed method, we observe a significant improvement in learning performance in simulated experiments as well as in real robots.
One of the arguments explaining this increase, presented in the paper, are the "compounding errors".
However, it is not directly clear how this conclusion follows from the observed improvements.
Perhaps, the origin of the increased performance could be subject of future investigations?


# Originality

The presented work combines known approaches to achieve an increase in learning performance.

From my perspective, two main contributions of the paper are:
- application of piecewise linear (constant) approximation in preprocessing for imitation learning
- experimental study of the effects of such approximation on learning performace

## Citations

Overall, the related work seems to be well cited and includes a number of recent publications.

## Algorithm: AWE

The authors propose an algorithm for "Automatic Waypoint Extraction" by approximation with continuous, piecewise linear functions.
There is an extensive body of work on approximation with piecewise linear (and piecewise constant) function in different disciplines.
It seems unlikely that the particular way of computing the approximation is completely novel.
Discussion / comparison of related approaches would substantiate the proposed method.
Besides the purely mathematical, related topics "Decision tree learning" and "Segmented regression" could be worth considering.

## Compounding Error

The only paper explicitely cited on "Compounding Error" is from 2011. Do other papers cited in section 2 explicitely analyze the reduction in "Compounding Error"?



# Significance

On the whole, the presented algorithm and results have a somewhat iterative character: the performance of an imitation algorithm is improved by an improved pre-preprocessing algorithm.
The observed improvements in learning, however, point to a potential for significant discoveries.

Especially promissing is the observed performance increase for a low number of training examples leading to a better data efficiency.
Low data efficiency is often a major hurdle for learning algorithms to be applied in real world scenarios.

More generally, perhaps, we might think of the way points as intermediate goals or events.
This line of research might contribute to a deeper understanding of the interplay between continous and discrete cognition.
Discrete actions and events seem to play a crucial role in human cognition.

An experimental study presented in [*] has shown that $L_0$ regularization for training
of recurrent neural networks can lead to improvements in learning performance. $L_0$
regularization enforces piece-wise constant changes in the latent state. Those sparse
changes are essentially events.

[*] Christian Gumbsch, Martin V. Butz, and Martius Georg. "Sparsely Changing
Latent States for Prediction and Planning in Partially Observable Domains"


# Relevance

The paper makes an empirical contribution to imitation learning and presents experiments performed on real robots:

- perform experiments in simulation and on real robots
- demonstrate improved performance in comparison to state of the art methods

As such, it is highly relevant to in the context of this conference.


# Limitations

The limitations-section discusses technical limitations of the proposed method.

In some experiments, the observed performance seems to decrease, as described in "Quality / " (Table 2, Figure 5).
This could point to potential limitation of the approach.

**Quality Of The Limitations Section:**

Limitations are addressed clearly

**Questions For Rebuttal:**

* is the actual interpolation linear or constant?
* does the augmentation with the closest waypoint serve as an intermediate goal and an outlook into the future (which might contribute to a more stable policy)?
* how can we derive that the reason for the improved performance is the reduction in "Compounding Error" based on the experiments presented in the paper?

**Robotics Focus:**

Sufficient demonstration on hardware

**Summary Of Paper:**


The authors propose an algorithm for "Automatic Waypoint Extraction", which approximates the demonstrated proprioceptive trajectory through a shorter sequence of "way points" to improve the performance of imitation learning algorithms by reducing the "compounding errors".

The points are selected by approximating the original trajectory with continuous piecewise linear / (piecewise constant) functions and added to the dataset as an additional modality.

The method is tested in a set of experiments in conjunction with two imitation learning techniques: Diffusion Policy (simulated benchmark) and Action Chunking with Transformers (ACT) (experiments with real robots).

The experiments show that learning methods perform significantly better with the approximated trajectories.

**Summary Of Recommendation:**

Overall good work, but clarity and quality could be impoved.
With minor improvements, this work can provide interesting insights and ground for fruitful discussion at the conference.

---

> ### Author Response · Authors · 2023-08-15
> **Any Further Clarifications?**
>
> Thank you once again for the extensive insightful comments! Please let us know if our response addresses your questions and whether there is additional clarification we can provide.

---

### Author Response · Authors · 2023-08-15
**Concern About Reviewer cGVi**

Respected AC,

We are concerned about the review from cGVi. While they raise some helpful points, the review contains factual misunderstandings and misunderstands the context in which the paper has been written, leading to misplaced concerns and suggestions.

> I think it’s success heavily relies on task domains and kind of demonstrations being used for training.

Our evaluation consists of two previously-proposed benchmarks with no modifications to the demonstrations or evaluation mechanism, as well as multiple real robot tasks. We believe the reviewer misunderstood the intuition behind the method, and that this statement is based on their inaccurate intuition and not the evidence to the contrary in the paper.

> waypoint that is close to the object state (this can be discovered directly from contact mode)

The reviewer incorrectly assumes that our method has direct access to extrinsic information about the environment (e.g. object poses, contact modes). This assumption is common in the motion planning literature, but our work is situated in an end-to-end robot learning context where we only have access to raw RGB images from the environment and proprioceptive information from the robot.

> we subsample trajectories that have been recorded at a high frequency we (i.e. reduce 100Hz trajectories to 10 Hz), we implicitly select waypoints and use linear interpolation between these waypoints

The paper already includes a direct comparison to uniform subsampling, which the reviewer seemed to miss. We also made this comparison more detailed in the revised version.

> The idea of using waypoints is not very novel.

The paper does not make this claim, and we clearly discuss prior works in the paper. We have also added a more detailed discussion of methods that use waypoints in the motion planning literature to the revised paper.

We would be grateful if you can account for these misunderstandings during the discussion and your final decision making process.

---

### Decision · Program_Chairs · 2023-08-30

**Decision:**

Accept (Poster)

**Comment:**

This paper proposes a method for imitation learning, which approximates demonstrated trajectories with a sequence of waypoints connected by linear paths. This decomposition is achieved by manually choosing an error budget, and then calculating the shortest subsequence of states that reconstructs the original demonstration within this error budget. Both simulation and real-world results show that, when combined with Diffusion Policy and Action Chunking with Transformers, training on these approximated trajectories significantly improves performance.

Reviewer scores before the rebuttal were: 1 x “weak reject” and 2 x “weak accept”. Authors provided a rebuttal, and an updated paper with some minor clarifications and additions. Following the rebuttal, the reviewer scores remained the same. The AC then initiated a discussion with the reviewers, focussing on the novelty and generality of the proposed method.

All three reviewers engaged in this discussion. There was not a general consensus here. Some reviewers argued that, whilst the idea of decomposing trajectories into waypoints is not novel, the particular way it is done in this method is novel. Other reviewers were concerned that the novelty was not significant compared to other methods proposing decomposition into waypoints. On the issue of generality, some reviewers argued that whilst the method may not work on more dynamic tasks and whilst there are hyperparameters to tune, the performance on static tasks is sufficiently strong. Other reviewers were more concerned that the method may not work well for more complex tasks, particularly those with multi-modal demonstrations.

Overall, despite the lack of consensus, I have a general feeling that the reviewers were more positive than negative. Personally, I appreciate the simplicity of this method and its strong real-world results. I believe that conceptually, it is an easy idea to grasp and discuss, and that the community will therefore benefit from hearing about this work. For the camera-ready paper, please be sure that the clarifications added since the rebuttal remain in the paper, because it is currently over 8 pages.